# ML-Based JIT1 Optimization for Throughput Maximization in Cluster Tool Automation

Youngsoo Kim [1], Gunwoo Lee [2] and Jongpil Jeong [1,*]

1    Department of Smart Factory Convergence, Sungkyunkwan University, 2066 Seobu-ro, Jangan-gu, Suwon 16419, Korea; young.kim@lamresearch.com
2    Lam Research Korea, Dongtan Sandan 5-Gil, Hwasung 18487, Korea; kelvin.lee@lamresearch.com
*    Correspondence: jpjeong@skku.edu; Tel.: +82-10-9700-6284 or +82-31-299-4267

**Abstract:** The semiconductor etch cluster facility is the most used facility platform in the semiconductor manufacturing process. Optimizing cluster facilities can depend on production schedules and can have a direct impact on productivity. According to the diversity of semiconductor processes, the complexity of optimization is also increasing. Various optimization methods have been studied in many papers for optimizing such a complex cluster facility. However, there is a lack of discussion of how these methods can apply to practical semiconductor manufacturing fabs and the actual performance results. Even now, data analysis and optimal parameter derivation for maximizing the productivity of cluster manufacturing in semiconductor manufacturing fabs are continuing. In this study, we propose an automated method for data collection and analysis of the cluster, which used to be done manually. In addition, the derivation of optimization parameters and application to facilities are addressed. This automated method could improve the manual analysis methods, such as simulation through data analysis using machine learning algorithms. It could also solve the inefficiency caused by manual analysis performed within the network inside the semiconductor manufacturing fabs.

**Keywords:** cluster tool; semiconductor manufacturing; throughput optimization; scheduling: ML-based data analysis; throughput prediction

## 1. Introduction

Research on the productivity of semiconductor cluster tools is being actively conducted in the increasingly diversified production environment of semiconductor manufacturing companies [1]. In semiconductor manufacturing, the single wafer processing method processes each wafer within a single facility and is widely used in cluster tools suitable for producing products that meet the various market requirements; it includes the lithography process, etching process, and deposition. It is also used in processes such as display manufacturing [2]. In particular, the quantity of semiconductor wafers produced per unit of time has become a field requiring more research. The importance of a foundry fab, a multi-variety environment, is becoming more prominent. Therefore, it is time to find a strategy to identify an optimization method for on-demand, up-to-date processes in existing cluster tools [3]. Technologies that can support services, through artificial intelligence, automation, optimization, and so on, will enable a direct or indirect effect not only on the production optimization of the already installed semiconductor manufacturing fabs but also on the optimization of semiconductor tools to be introduced in the future [4].

The scheduling of equipment units can be divided into deterministic and unstable environments. A deterministic environment mathematically performs the scheduling without considering arbitrary elements inside the equipment [3]. In a variable environment, a mathematical approach is not easy to perform, and an approach considering various situations and conditions inside and outside the facility is required. The research method under a deterministic environment must explain the productivity results of cluster tools that are

changed by various variables in the real world, so there are limitations in the practicality of studies [5]. Field engineers are also trying to analyze and improve the throughput affected by these various variables [6]. Recently, different situations have been examined, such as when several types of wafers are processed in parallel, when a failure occurs during the process, or when the chamber schedule is unstable [4,7].

To date, when throughput problems occur, the cause of the problem has been found in the following way. First, correlation analysis has been done between the measured values of equipment and throughput. Second, throughput correlation analysis has been done through abnormal operation analysis of the facility controller. Third, data comparison analysis has been performed through simulations. Recently, throughput modeling through data analysis using various ML methods has been proposed, but several points should be considered as follows. Scheduling strategies differ because data analysis based on real-time data is not stable. Therefore, an integrated schedule model, analysis method, and real-time facility optimization method has not been presented [4,7]. The use of unoptimized equipment may not result in optimal throughput results [8]. To simultaneously solve these two problems, some researchers have modeled the initial transition state schedule with Petri net and used operational research methodologies such as linear algebra or MIP (mixed-integer program). A study was conducted to determine the work's starting point [9,10]. Other researchers have tried to show that the best method is to use a strategy in which a possible task starts late in the start-up transient period and a strategy in which a possible task starts in a short end period. In addition, a PERT chart has been used. However, since these studies proposed a scheduling strategy considering the constant working hours of robots and PMs within the initial transition period, it is not clear whether the constraint can be satisfied even if there is time fluctuation [4,7,11].

In this study, we present an ML-based JIT1 optimization strategy using real-time data, robust to real-time fluctuations on cluster equipment. To date, various strategies have been studied to improve semiconductor tools' efficiency [12]. In other words, there have been studies to improve throughput using simulation and compare strategies' performance with changed schedules [11]. This study is based on the schedule change of the current initial transition period. We propose a strategy to respond to the time fluctuation of the optimal initial transition in the stable state so that it can be connected to the schedule of the ideal stable state. In addition, simulations are performed to verify these strategies and collect data. The real-time collected data allowed optimization modeling of cluster tools suitable for a specific process through ML analysis, and simulations were performed to verify these strategies. The main contents of this paper are as follows.

- To understand the structural characteristics of cluster tools, ML-based analysis of JIT1 is proposed to maximize throughput, and an automation system is used to predict and apply optimal JIT1 to facilities. Lam Research's cluster facility simulation was used to obtain cluster facility data, and Lam Research's host simulation was used for automation modeling. The simulation configuration cited the SECS/GEM procedure of semiconductor manufacturing fabs.
- Various data analyses were performed to maximize throughput in the semiconductor equipment field, but the efficiency was lowered because the analysis was not automated. In order to solve this problem, an automated technique was used to obtain the optimal JIT1 with an ML-based analysis method. This process shortened the time-consuming data analysis process and, at the same time, guaranteed maximum throughput while sensitively responding to changes in the production environment in real time by applying an automation system.

The structure of this paper is as follows. First, in Section 2, cluster equipment is introduced. After discussing the operation method of cluster equipment, various factors affecting the throughput of cluster equipment are explained. Moreover, the research related to throughput maximization that has been conducted so far is addressed. Moreover, in Section 3, JIT1 optimization and real-time cluster facility optimization process methods through ML-based analysis for maximizing throughput based on real-time data of cluster

tools are described. Section 4 compares and evaluates the strategies proposed through the automated simulation proposed in Section 3. Section 5 discusses the results, implications, and future tasks of this research.

## 2. Related Work

### 2.1. Structure and Operation of Cluster Tools

The cluster facility is a single wafer processing automation system widely used in semiconductor wafer processes. It is used not only for the etch facility, which is the research topic of this paper, but also for the deposition and exposure tools [13]. It comprises one cluster or several process modules (PM), and wafer processes can be executed in parallel. Figure 1 shows a cluster facility consisting of three process modules, two airlocks, a transfer robot with two arms, and three FOUPs [14].

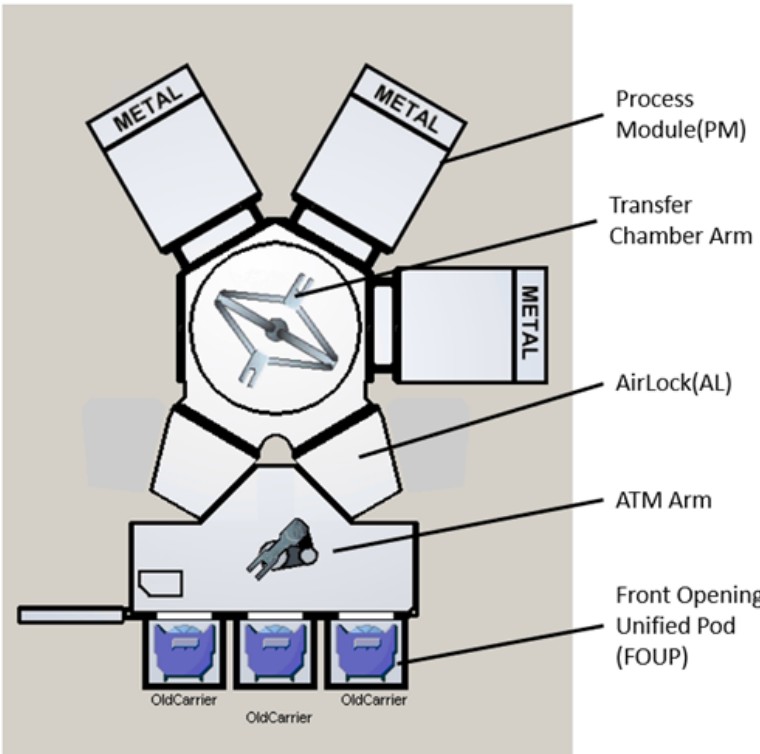

**Figure 1.** Cluster facilities including three PM, two AL, and three FOUP.

The initial wafer is stored in the FOUP, and the OHT of the semiconductor fab transports the FOUP and moves it to the load port of the facility. For the wafer that has finished mapping at the load port, the ATM arm returns it from the load port to the AL, and the transfer chamber arm selects one of the processable PMs from the AL and returns the wafer. The wafer arrives at the PM and proceeds with the pre-programmed recipe process [15]. When the wafer's recipe process is completed, through the transfer chamber arm again, it moves from PM to AL. Then, the ATM arm transfers it from AL to FOUP, and OHT retrieves the FOUP. After completing the process, the PM proceeds with the WAC process, and a chamber-cleaning process maintains the quality required for the next process in the recipe [7]. Suppose the cluster facility uses the clean recipe. In that case, the PM must proceed with the removal process of residual chemicals and impurities formed inside the PM for the subsequent wafer processes, a process called WAC time. After this WAC time, subsequent wafers can move to the PM [5]. The schedule of the cluster facility is determined according to recipe time, robot transmission time, and availability of PM and AL [16], as well as the above OHT return and wafer map verification process recipe download and

other parameters. These are carried out through communication based on the SECS/GEM communication protocol with the host of the semiconductor fab.

### 2.2. Productivity Parameters of Cluster Tools

JIT1 is one of the scheduling variables of cluster facilities, referring to the time when the wafer moves from the FOUP. For example, if JIT1 is N, the scheduler starts moving the wafer from the FOUP to airlock N seconds before. Depending on this time point, the total moving time of the wafer may change. The PM is chamber space in which the semiconductor wafer recipe operates as a process module. Process time is the processing time of wafers. WAC time is an abbreviation for waferless auto-clean, referring to the cleaning time of the chamber space for the following wafer process.

Wafers stored in the FOUP are in the initial status, and the OHT of the semiconductor fab transports the FOUP and moves it to the facility's load port. After wafer mapping on the loadport, the ATM arm returns it from the loadport to the AL, and the transfer chamber arm selects one of the processable PMs from the AL and returns the wafer [17]. The wafer arriving at the PM proceeds with the pre-programmed recipe process. When the wafer's recipe process is completed, again through the transfer chamber arm, it moves from PM to AL. Then, the ATM arm transfers it from the AL to FOUP, and the OHT retrieves the FOUP. After completing the process, the PM proceeds with the WAC process, a chamber-cleaning process to maintain the quality required for the following process in the recipe [18]. Suppose the cluster facility uses the clean recipe; in this case, the PM must proceed with removing residual chemicals and impurities formed inside the PM for the subsequent wafer processes, a process called WAC time. After this WAC time, subsequent wafers can move to PM. According to the recipe time, the schedule of the cluster facility, the robot's transmission time, and the availability of the PM and AL are determined. The above OHT return and wafer map verification process download and upload the recipe. These are carried out through communication based on the SECS/GEM communication protocol with the host of the semiconductor fab.

### 2.3. Maximizing the Throughput of the Cluster Tool

Because the number of chambers and robots in the cluster facility is limited and repetitive tasks are performed, depending on the scheduling problem, throughput can be variable, which seems to be simple in a limited environment due to the characteristics of cluster tools. Based on facility variables, throughput has a non-linear relationship, so optimization to perform optimal scheduling is not simple [19–22]. The throughput of the cluster tool is determined according to the throughput of the number of wafers that have completed the specified process recipe for a unit of time (1 h). Throughput is the most important variable to obtain maximum productivity of cluster tools because the scheduling is changed according to the production plan of the semiconductor fab and the setting value of JIT1. In addition, throughput determines the production system quality, which must satisfy the user's needs [23,24]. However, it is not easy to predict the sustainable throughput of cluster tools due to the complex scheduling and unexpected real-time problems. To date, extensive research has been conducted on the scheduling and throughput of cluster tools [13,25,26]. Various throughput models have been studied according to various cluster facility configurations and under conditions such as sequential or parallel PM process or WAC [1,21,27]. However, this model has a limitation in providing the theoretical throughput of a specific cluster facility. In reality, cluster tool modeling may not follow theoretical modeling due to various real-time problems. Therefore, an optimization method using a more practical throughput model is needed [21]. Although this timing-based model is easy to understand, it has a limitation since it cannot effectively consider the theoretical throughput of the cluster facility, that is, the real-time scheduling problem. Therefore, a more realistic Petri net modeling study was conducted [28].

Throughput (WPH—wafer per hour), which indicates the number of wafers produced per unit time of the cluster facility, depends on JIT1 and the number of PMs, recipe time,

and WAC time. Collecting the throughput results measured under various parameter conditions is essential to obtain modeling for accurate throughput by understanding the relationship between each variable and throughput. This operation is very time consuming, but using big data, it can be possible in semiconductor fabs. However, this study used different values for the four main parameters, such as the number of PMs, recipe time, WAC time, and process time, while 23 values were used for JIT1. In addition, since the data of the semiconductor fab cannot be used for security reasons, as in Figure 2, the throughput result data by simulation of Lam Research were used. Lam Research is a cluster facility based on the Petri net model. Petri net configuration simultaneously processes the loops of PM, AL, and FOUP [29,30].

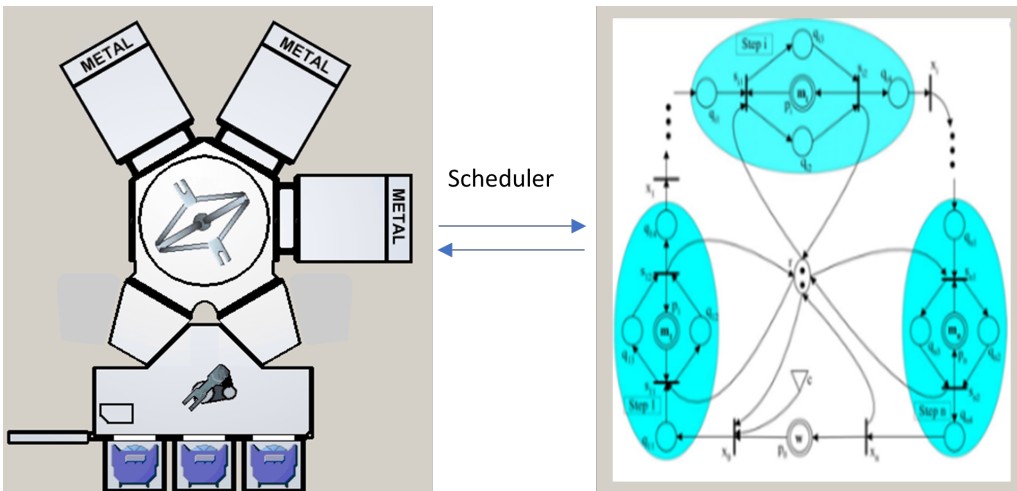

**Figure 2.** Petri net model of cluster tool simulation.

The data of the cluster facility is standardized and stores each robot's controller action execution event. WPH is calculated through wafer movement time, wafer process time, wafer WAC time, and waiting time. After the production of the unit, the lot is finished, and the result is saved as an event. This can be expressed as Equation (1) for the calculation of WPH [30].

Equation for maximum productivity of cluster facilities:

$$Throughput = \frac{(Number\_of\_Chamber\_ \times 3600 \text{ s/h})}{(Process\_Time + WAC\_Time + Wafer\_Swap\_Time)} \quad (1)$$

Here, the wafer swap time is a variable that calculates the process rate from when the wafers starting from the load port are out of load in all chambers until all wafers are returned to the load port. Throughput calculated in this way is immediately saved as event data after the entire lot process is finished. Therefore, it is possible to find direct and indirect influence factors by analyzing variables related to throughput through event data analysis [31].

Figure 3 shows the relations between process time, WAC time, JIT1, throughput, PM, and target. PM means how many process modules are used. Target refers to calculated result data from the equation for maximum productivity of cluster tools. There is a significant correlation that could not be observed between throughput and JIT1.

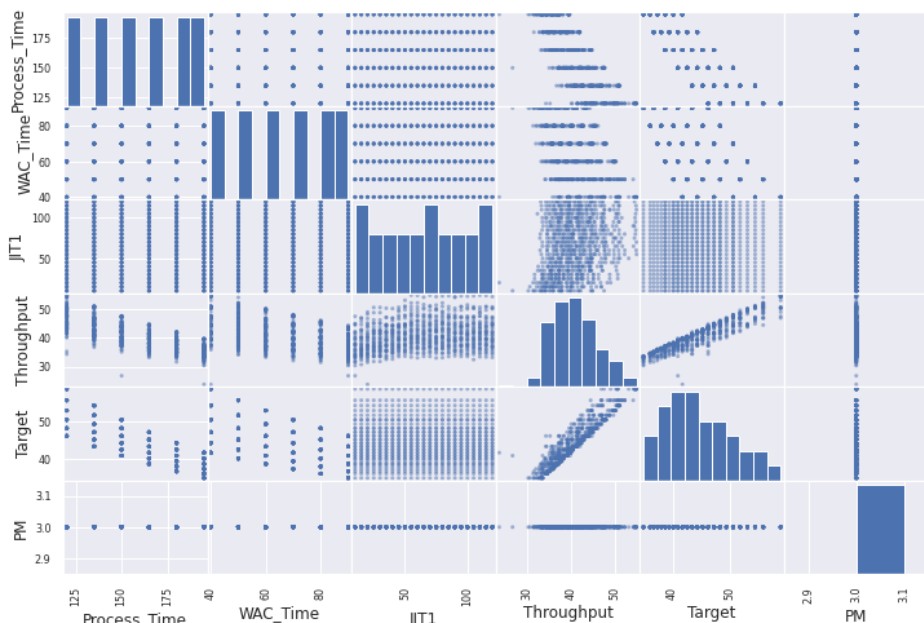

**Figure 3.** Correlation between throughput and process time, WAC time, and JIT1.

Figure 4 shows that the following steps should be followed to optimize JIT1 for the current cluster facility. A data collection process is required for data analysis, but the data has to be processed due to security concerns in the semiconductor manufacturing fab. It takes significant time to analyze and simulate the processed data [7]. Manual simulation implementation can be inefficient because it takes a similar amount of time as the primary setting of the cluster facility. In addition, application through simulation is also conducted passively, so the above analysis method is a continuous inefficient work method under the multi-variety production system.

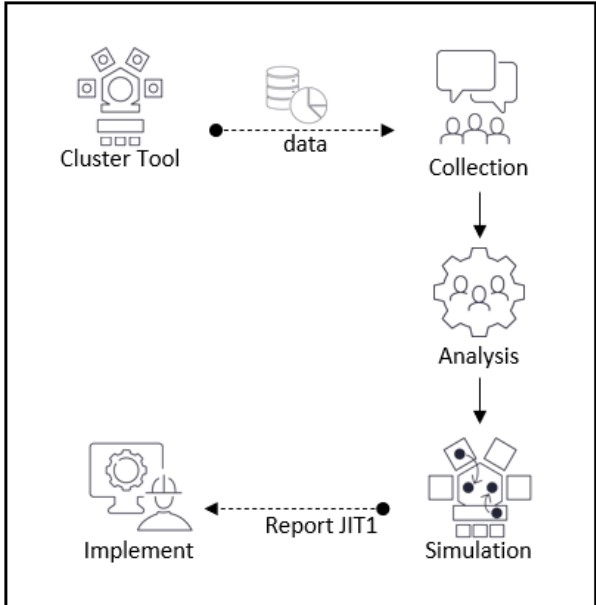

**Figure 4.** JIT1 optimization of semiconductor cluster tool (current).

## 3. JIT1 Optimization for Throughput Maximization

The throughput maximization method of semiconductor etch cluster facilities proposed in this paper analyzes the current performance of the cluster through ML-based

algorithms, predicting the optimal JIT1, and reflecting real-time optimization parameters for the cluster. The semiconductor etch cluster facility can be changed according to mechanical factors such as load port performance, transfer robot performance, airlock pump/vent performance, VTM robot performance, and various door performances. In addition, characteristics of the semiconductor etching process and the process of the chamber may be changed depending on the process time, when the wafer performs the process in the chamber, and the WAC time, when the wafer proceeds without the wafer in the chamber. However, the method to obtain maximum productivity considering the constant mechanical performance, process time, and WAC time of this etch cluster lies in the optimization of JIT1 parameters related to scheduling that affect the start time of wafer movement [19]. The optimization of JIT1 can be obtained through simulation modeling of various parameters by analyzing the data of the cluster. However, this modeling simulation method is very inefficient in terms of time. The method proposed in this study is shown in Figure 5. We designed an architecture to optimize productivity by combining the optimal algorithms for data analysis of a semiconductor cluster through data generated from the cluster. KNN is the optimal ML algorithm for a semiconductor etch cluster [4]. The facilities of the semiconductor manufacturing fab are ready for daily 24-h production, and data generated from the facility can be collected in real time as standardized data. The content in the standardized data includes information such as process time, WAC time, and wafer transfer time of the wafer being processed in the facility. WPH can be analyzed through this.

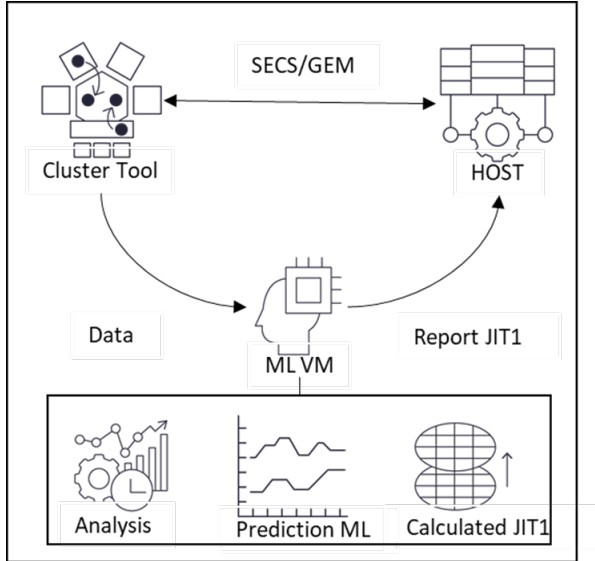

**Figure 5.** JIT1 optimization of semiconductor cluster tool (proposal).

*Throughput Optimization through ML*

Studies on data analysis and prediction methods using ML VM, as shown in Figure 5, are already in progress in various fields. However, this paper studied an efficient method of using ML systems in a semi-standard environment. This study confirmed the possibility of real-time data analysis and productivity improvement. Throughput data generated by a cluster tool is transfer to the VM and analyzed using ML. As a result of analysis and prediction using ML, the optimal JIT1 value is reported to the host, and the host commands a change in the JIT1 parameter of the cluster tool when producing the following production lot. This is a way to maximize productivity through more efficient work by presenting cases in which optimal parameters can be predicted and applied through data analysis using ML-based algorithms in the fab, considering the characteristics of a semiconductor fab with strict security. The productivity analysis of the cluster includes the data collection and analysis described in Section 3. The analyzed data manually create a test scenario including process time, WAC time, swap time, and JIT1.

In Table 1, you can see a process scenario and the simulation result. In this paper, we will present an automated method through ML analysis based on the dataset of the cluster facility through simulation and a method to obtain the maximum productivity of the semiconductor etch cluster facility through a real-time analysis and application process.

The following methods are used to improve the throughput of the etch cluster facility of the semiconductor manufacturing fab. The first method classifies and analyzes the motion data of various components generated in the cluster to create and reproduce simulation scenarios or compare and analyze scenario simulation data and field data to find parameters that need improvement and apply them to facilities to identify problems and solve them. The second method calculates the maximum throughput under the current facility configuration and sets it as a target, compares it with the facility's data, finds the problem, finds the variable to be improved, and applies it to the facility. Such comparative analysis requires significant time and effort through copying and moving data, replicating simulators, and post-comparison. In addition, the application of variables in the cluster is a manual variable adjustment method used by engineers. It is possible to first analyze the log generated in the equipment to change the JIT1 to obtain the maximum productivity of the equipment in the current semiconductor manufacturing fab. Through this, it is possible to obtain maximum efficiency with the existing method.

**Table 1.** Cluster tool throughput simulation data.

| Process Time | WAC Time | Tput (JIT = 10) | Tput (JIT = 15) | ...... |
|---|---|---|---|---|
| 120.00 | 40.00 | 47.51 | 46.92 | ...... |
| 120.00 | 50.00 | 44.48 | 46.87 | ...... |
| 120.00 | 60.00 | 44.85 | 44.59 | ...... |
| 120.00 | 70.00 | 41.62 | 42.38 | ...... |
| 120.00 | 80.00 | 40.14 | 41.50 | ...... |
| 120.00 | 90.00 | 35.29 | 34.37 | ...... |
| 135.00 | 40.00 | 44.91 | 45.78 | ...... |
| 135.00 | 50.00 | 42.24 | 42.69 | ...... |
| 135.00 | 60.00 | 40.81 | 40.12 | ...... |
| 135.00 | 70.00 | 39.13 | 39.97 | ...... |
| 135.00 | 80.00 | 38.04 | 39.09 | ...... |
| 135.00 | 90.00 | 37.39 | 36.76 | ...... |
| 150.00 | 40.00 | 41.88 | 42.95 | ...... |
| 150.00 | 50.00 | 39.61 | 40.08 | ...... |
| 150.00 | 60.00 | 37.24 | 37.51 | ...... |
| 150.00 | 70.00 | 37.88 | 37.89 | ...... |
| 150.00 | 80.00 | 35.94 | 37.10 | ...... |
| 150.00 | 90.00 | 26.81 | 35.11 | ...... |
| ...... | ...... | ...... | ...... | ...... |

The method proposed in this paper is to automate the data collection, data analysis, and deduction of variables, to obtain maximum productivity and reflection in JOB production planning. The data collection uses the event-based file transfer system to transfer files to the VM when they are created. In addition, data analysis uses an ML-based analysis algorithm based on the accumulated data. Figure 6 shows the data analysis result based on the KNN algorithm as a graph. Based on this, it is possible to obtain the optimal JIT1 value under various production environments. JIT1 derived in this way is applied to the facility through the host.

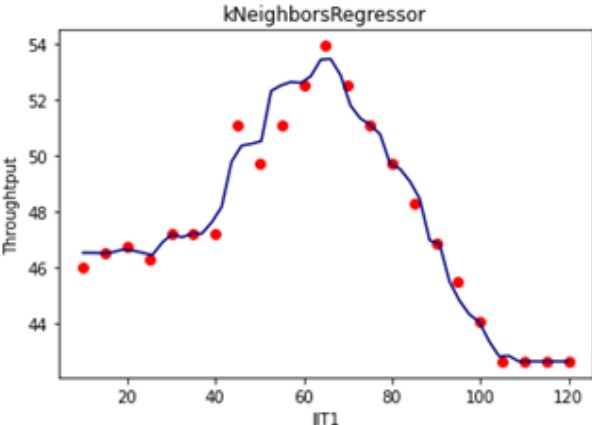

**Figure 6.** Throughput prediction through KNN regression (process_time = 135, WAC_time = 40).

### 4. Simulation and Results

This paper used a simulation of Lam Research to propose a more efficient data analysis method in an environment under an increasingly strengthened security system in the semiconductor industry. Lam Research manufactures various facilities for the semiconductor process, including for deposition, cleaning, and etching. Herein, we aimed to study the productivity optimization method based on mutual communication of SECS/GEM protocol between the etch cluster facility and the fab host based on the actual scenario used in the etch cluster semiconductor production facility. Figure 7 is a schematic diagram of the simulated layout of Lam Research. The simulation of Lam Research made it possible to build an environment similar to the existing facility by simulating all IO and setting values of the existing facility. Based on the experience of actual facility operation, the user can build the communication environment of PMC and TM as well as CTC and TMC internally and externally with the factory host. The CTC acts as central control, collecting and generating data. Through communication between the factory host and CTC, process progress and real-time commands transmitted from the host are executed.

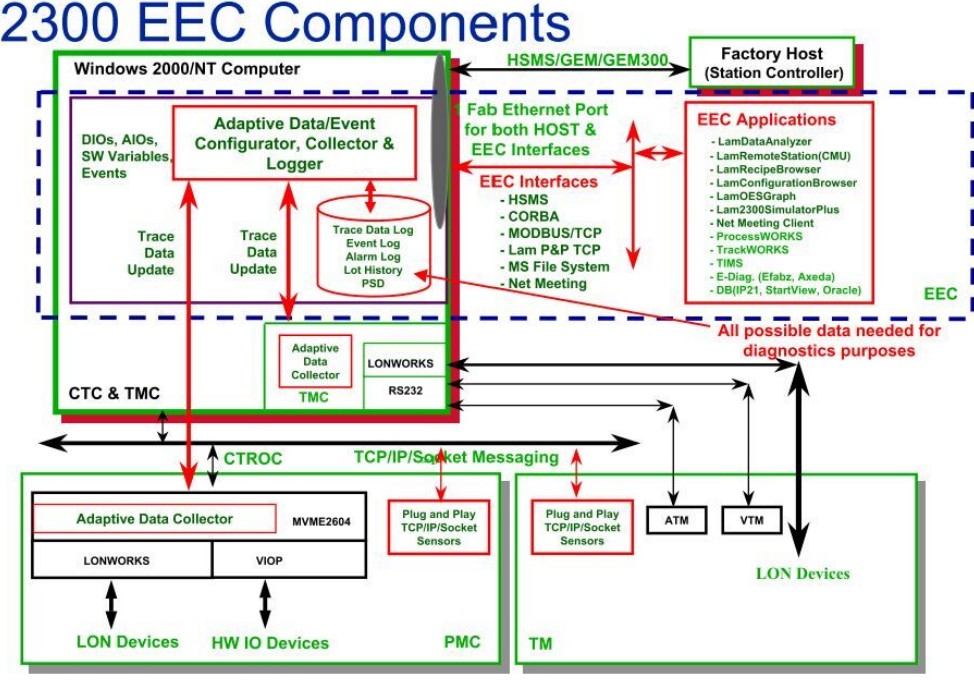

**Figure 7.** Lam Research simulation architecture.

### 4.1. Simulation Environment

Due to the nature of the semiconductor manufacturing fab, the export of internal data is extremely limited per regulations, such as those regarding data security. Therefore, the study of this paper used a simulation of Lam Research, a semiconductor manufacturing equipment company. The simulation of Lam Research can create an environment similar to the existing facilities by using a variety of changeable parameters. Since each component can set the allowable interval for temperature, pressure, and other parameters, the simulation settings were set similarly to the environment of the facilities in the semiconductor fab. It can be said that it is functional, as shown in Figure 7, and it was possible to change the DIOs, AIOs, and characteristics according to the SW version of the facility. In addition, it was possible to change not only the process time and WAC time of the wafer process but also the transfer time of the ATM robot and VTM robot and the pump/vent time of the airlock. Therefore, it was possible to reproduce the performance of the existing facility by setting it similarly to the environment of the etch cluster facility in the semiconductor manufacturing fab. Additionally, the SECS/GEM communication environment between host and cluster equipment can be simulated. This simulation scenario can be applied to various settings of process time and WAC time by predicting the optimal JIT1 according to process time and WAC time using the ML-based KNN algorithm suggested in the previous study and applying it to the real-time etch cluster facility. This simulation environment provides a highly reliable test environment in which the throughput result is kept constant if no parameter changes. Therefore, the optimal JIT1 value is determined according to the throughput result. Various commands of factory host can be executed, and it is possible not only to collect/transmit real-time data but also to check and verify recipes.

### 4.2. Cluster Tool Control in Semiconductor Manufacturing Fab

Etch clusters are operated based on various parameters. This paper proposes two strategies: the ML modeling method that could obtain the optimal JIT1 based on process time and WAC time, and the modeling that could obtain the maximized throughput by applying the optimal JIT1 to the real-time cluster facility. In general, in the case of the cluster, it was difficult to optimize throughput according to the real-time changing process time and WAC time because JIT1, to obtain maximum throughput, was used as a fixed value instead of being variable in real time. Therefore, modeling to obtain optimal JIT1 through the ML analysis was proposed. A strategy was presented to apply JIT1 to a cluster in semiconductor factories along with lot allocation through the SECS/GEM protocol. The productivity optimization process proposed in Section 3 consisted of simulations. Figure 8 shows the simulations of the cluster facility being controlled by the host. The two on the right give various commands to the equipment through the host and receive real-time data. The one on the left is a cluster facility, and when it receives a command from the host, it executes the command. These commands included the progress of the lot and the change of parameters such as stop and recovery. In addition, real-time process parameters could be changed, ensuring versatile plant control.

### 4.3. Result

As can be seen from the relationship between JIT1 and throughput in column 4 and row 3 in Figure 3, a significant correlation could not be observed between throughput and JIT1. However, as shown in Figure 9, it was confirmed that the throughput showed a significant correlation with JIT1 when the processing time was fixed at 180 and the WAC and other parameters of JIT1 were changed. In Figure 9, it can be seen that the throughput had an S curve, but the point at which the throughput was stabilized differed depending on the WAC time and JIT1. Therefore, it can be seen that for optimizing the cluster, it is necessary to set the optimal JIT1 considering the processing time and WAC time. There are several ways to find the optimal JIT1 under the variable environment of process time and WAC time. We can determine whether the cluster facility operates efficiently through

Equation (1). If the cluster facility's throughput is smaller than the calculation result of Figure 9, the facility is operating at a non-rate.

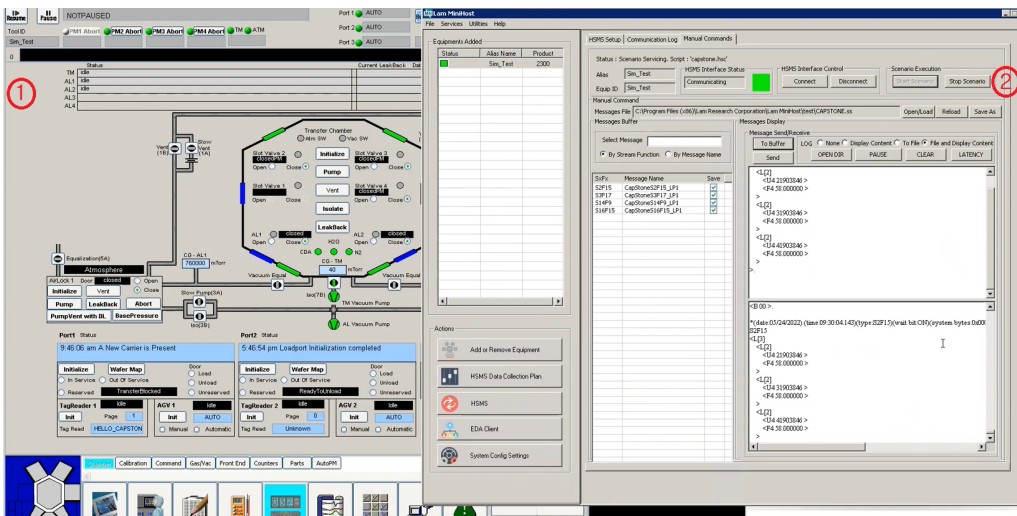

**Figure 8.** Etch cluster tool simulation connected to semiconductor fab host.

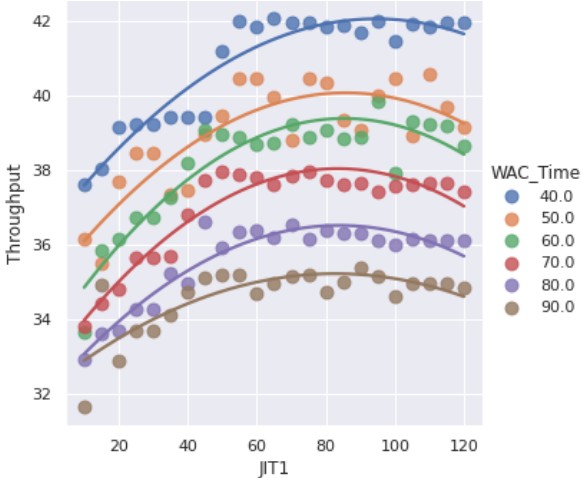

**Figure 9.** Throughput against JIT1 with different WAC time (recipe_time = 180).

Table 2 shows the Pearson coefficient and RMSE results of ML modeling. The Pearson coefficient of linear modeling was 0.941, and RMSE was 6.8, so it can be said that it was unsuitable for throughput analysis of the cluster. The optimal algorithm for ML modeling of the cluster was KNN, which had a Pearson coefficient of 0.989 and an RMSE of 3.28.

**Table 2.** Pearson coefficient and RMSE.

| Models | Pearson Coefficient | RMSE |
|---|---|---|
| Linear regression | 0.941 | 6.80 |
| Polynomial regression | 0.949 | 6.32 |
| SVM with rbf | 0.974 | 4.76 |
| KNN | 0.989 | 3.28 |

As a result of the study, it was possible to obtain a significant throughput improvement result in the facility, in which JIT1 optimization through simulation was applied. It can be assumed that similar results can be obtained in actual semiconductor fabs. This study confirms that it is possible to predict and apply optimization parameters by analyzing the data generated from the etch cluster facility in the semiconductor fab using the ML-based algorithm in the fab.

Figure 10 is the result of working to achieve maximum productivity through the methodology presented herein. You can check the error of the target by comparing it with the engineer's manual analysis and result derivation process. Through simulation, you can check the throughput result. The host's command to change the value of JIT1 to 58 is accepted, and the simulator is shown in Figure 10. The simulation result of Table 3 can be obtained. The value of JIT1, before being modified to 58, was set to 30, and without the modification, it showed a throughput of 37.06. However, by modifying to 58, it showed a throughput of 38.20, and it was confirmed that it would have better productivity than before. The calculated maximum value is an idealized maximum throughput value calculated by Equation (1).

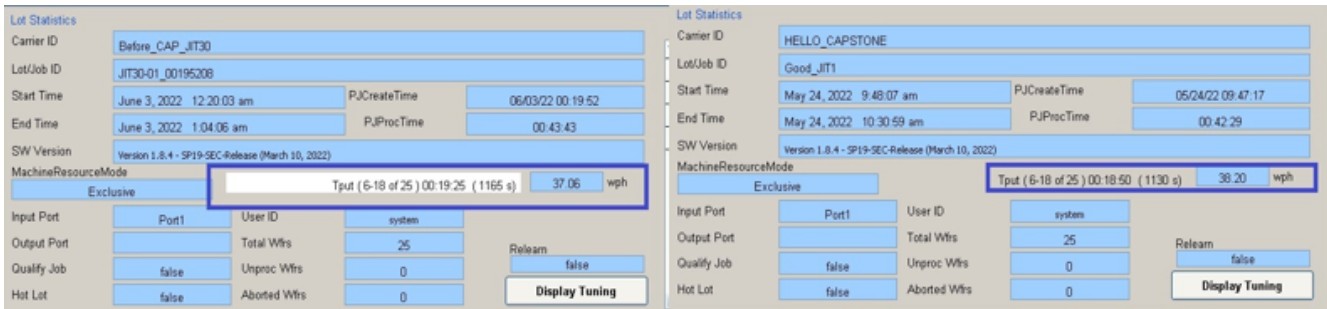

**Figure 10.** Throughput comparison of JIT1, before (**left**) = 37.06 and after (**right**) = 38.20.

**Table 3.** Throughput comparison of optimization.

| Process Time | WAC Time | Tput (Pre-Optimization: JIT1 = 30) | Tput (Post-Optimization: JIT1 = 58) | Tput (Caculated Maximum) |
|---|---|---|---|---|
| 195.00 | 50.00 | 37.06 | 38.20 | 40.14 |

In this section, we propose the utility of JIT1 optimization using ML-based data analysis using the KNN algorithm. When the user inputs process time and WAC time using the chatbot, the optimal JIT1 value is given. Through this chatbot service, the user can know the appropriate JIT1 according to the equipment's various products, that is, the products of variable process time and WAC time, and can apply it to the equipment. Figure 10 shows the result of improving the productivity of the cluster facility by changing the optimal JIT1 obtained by the user through the KNN ML analysis through the host. The chatbot service is shown on the right side of Figure 11. Implementing the chatbot in the simulation environment can inform users of the optimal JIT1 value according to various process times and WAC times in real time and be an appropriate alternative to the increasingly diversified production environment.

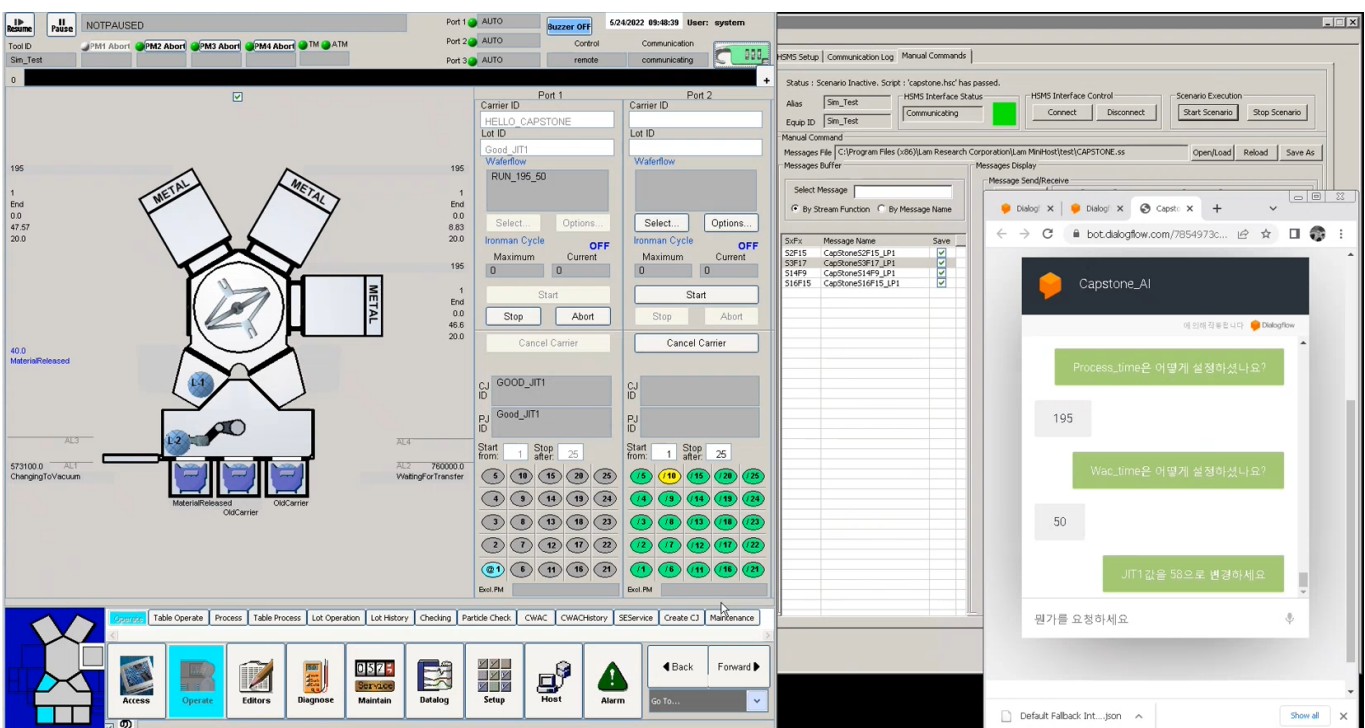

**Figure 11.** Optimal JIT1 value using chatbot utility.

## 5. Conclusions

As a significant contribution of this paper, productivity analysis and a JIT1 optimization progress flow model of a semiconductor cluster are proposed for data analysis and optimization methods in semiconductor fabs where data security is essential. Partial automation is possible because there are many manual elements for each step in the existing process model, but it was challenging to find an overall study of the process automation model. This paper introduced machine learning to data analysis to derive the optimal JIT1 and report it to the host, and a simulation result of real-time optimization was obtained. Therefore, fast real-time optimization of the semiconductor etch cluster can be guaranteed. Low latency, high reliability, and high availability can be ensured as a system advantage at the semiconductor manufacturing site. In the future, it will be necessary to analyze the optimal ML modeling of the extended various cluster tools' configuration parameters, which can expect maximum throughput in the configuration environment with various cluster chambers, including deposition and ashing, as well as the etch cluster facility; and to study the production method strategy. Furthermore, the optimal algorithm that analyzes various data generated from the etch cluster facility can be a research subject. Data analysis in such an unstable environment is meaningful in analyzing various parameters that directly affect a semiconductor process's productivity (i.e., yield).

**Author Contributions:** Conceptualization, Y.K.; software, G.L.; simulation, Y.K.; writing—original draft preparation, G.L.; writing—review and editing, J.J.; simulation, G.L.; supervision, J.J.; project administration, J.J. All authors have read and agreed to the published version of the manuscript.

**Funding:** This research was supported by the MSIT (Ministry of Science and ICT), Korea, under the ITRC (Information Technology Research Center) support program (IITP-2022-2018-0-01417) supervised by the IITP (Institute for Information & Communications Technology Planning & Evaluation). Also, this work was supported by the National Research Foundation of Korea (NRF) grant funded by the Korea government (MSIT) (No. 2021R1F1A1060054).

**Institutional Review Board Statement:** Not applicable.

**Informed Consent Statement:** Not applicable.

**Data Availability Statement:** Not applicable.

**Conflicts of Interest:** The authors declare no conflict of interest.

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
