# Peer review of "ML-Based JIT1 Optimization for Throughput Maximization in Cluster Tool Automation"

_applsci, doi:10.3390/app12157519_

Round 1
Reviewer 1 Report
This paper discusses the throughput improvement of cluster facilities with activity time variations in semiconductor manufacturing. It focuses on the etching tools in wafer fabrication. The topic is interesting. The authors claim that they propose an ML-based automated method to optimize the so called JIT1 so as to maximize the throughput. However, the paper is not properly presented.
1. The main concern is that it does not present a method and it just states as “The method proposed in this paper is to automate data collection, data analysis, deduction of variables to obtain maximum productivity, and reflection in JOB production planning.” In fact, totally it uses just a paragraph to state the method. This makes no sense.
2. Without presenting the method in any detail, the simulation results are not convincing.
3. The paper is poorly presented
1) It presents no formal description for the problem and the method.
2) Generally, figures are very useful for a scientific paper. This paper presents a number of figures. However, it gives no explanations for most of the figures, which makes the figures not meaningful.
3) Some statements are not understandable, for example, “Fig. As in ??” in Line 169 and “In 2” in Line 228 are such statements.
4. The language is poor and is not acceptable.
Author Response
We thank you for your time and consideration of our submission (Manuscript ID: applsci-1793568). Attached we address the referee's comments and a list of changes we made to our manuscript according to their reports. The appropriate changes made in the revised manuscript are highlighted.
We believe these modifications have strengthened the manuscript and hope the revised manuscript is suitable for publication in Applied Sciences.
Sincerely.
Youngsoo Kim

Reviewer 2 Report
The authors study the study of “ML Based JIT1 Optimization for Throughput Maximization in Cluster Tool Automation».
1. The authors should give the motivation of Optimization of cluster in introduction.
2. The authors should cite these references in introduction section theses references:
DOI 10.1007/s12633-021-01374-z ; DOI 10.1007/s12633-019-00214-5.
3. I purpose to the authors to give the Table 1 in text form.
4. The authors should explain more the method used in this work.
5. The authors should give the parameters obtained by optimization and those used in their calculation in one table.
6. The authors should give the recent refs.
In general, the manuscript has a correct methodological structure. But for my opinion the authors should making a major revision on this manuscript. I hope that authors reconsider these all points and they clarify some issues.
Author Response

(The authors gave the same response as above.)

Round 2
Reviewer 1 Report
Improvements are made by the revised version and more improvements can be made.
1. It may be useful to give more detail about the proposed method.
2. In Reference 28, the order of the authors and the publishing year are not correct.
Author Response
Dear Reviewer,
Please see the attached for your review and comments.
Best Regards,
YoungsooKim

Reviewer 2 Report
The authors are improved their manuscript.
Author Response

(The authors gave the same response as above.)
